# The associations between thermal variety and health: Implications for space heating energy use

**Harry R. Kennard** *, Gesche M. Huebner, David Shipworth, Tadj Oreszczyn

Energy Institute, UCL, London, United Kingdom

* h.kennard@ucl.ac.uk

## Abstract

Fossil fuels dominate domestic heating in temperate climates. In the EU, domestic space heating accounts for around 20% of final energy demand. Reducing domestic demand temperatures would reduce energy demand. However, cold exposure has been shown to be associated with adverse health conditions. Using an observational dataset of 77,762 UK Biobank participants, we examine the standard deviation of experienced temperature (named here *thermal variety*) measured by a wrist worn activity and temperature monitor. After controlling for covariates such as age, activity level and obesity, we show that thermal variety is 0.15°C 95% CI [0.07–0.23] higher for participants whose health satisfaction was 'extremely happy' compared to 'extremely unhappy'. Higher thermal variety is also associated with a lower risk of having morbidities related to excess winter deaths. We argue that significant $CO_2$ savings would be made by increasing thermal variety and reducing domestic demand temperatures in the healthiest homes. However, great care is needed to avoid secondary health impacts due to mould and damp. Vulnerable households should receive increased attention.

## Introduction

In the temperate climates of the Northern Hemisphere, domestic energy demand is dominated by space heating. In the USA there are four times as many heating degree days as cooling degree days [1]. In the EU, domestic space heating accounts for 78% of domestic energy use, at least 60% of which comes directly from fossil fuel sources [2]. A sensitivity analysis of the Cambridge Housing Model for the UK government [3] estimated that a 1°C drop in demand temperature decreased $CO_2$ emissions by 13%, making heating demand temperature one of the behavioural factors with the highest potential impact on emissions. Therefore, given the necessity of clear action on carbon emissions, reduction in domestic heating demand is vital.

At the same time, there is broad epidemiological consensus that observed seasonal variations in mortality in temperate countries is attributable to cold external temperatures and cold exposure. A recent meta-review of existing systematic reviews concluded that cold exposure and cold spells increase the risk of cardiovascular and respiratory illness and mortality [4]. Low temperatures in particular are known to exacerbate respiratory health conditions such as

**Data Availability Statement:** Restrictions limit the direct sharing of the data used in this study. However, the data used in this study are available to any bona fide researchers following application to a third party, UK Biobank at https://www.

ukbiobank.ac.uk/register-apply/. Access queries may be directed to access@ukbiobank.ac.uk. All data were fully anonymised by UK Biobank prior to being shared with the study authors.

**Funding:** HRK: Engineering and Physical Sciences Research Council Centre for Doctoral Training in Energy Demand (LoLo), grant numbers EP/L01517X/1 and EP/H009612/1. http://www.lolo.ac.uk/ GMH and HRK: UK Research and Innovation through the Centre for Research into Energy Demand Solutions, grant reference number EP/R 035288/1 https://www.creds.ac.uk/ GMH and HRK: Research Councils UK Centre for Energy Epidemiology EP/K011839 https://gow.epsrc.ukri.org/NGBOViewGrant.aspx?GrantRef=EP/K011839/1 The funders had no role in study design, data collection and analysis, decision to publish, or preparation of the manuscript.

**Competing interests:** NO authors have competing interests

chronic obstructive pulmonary disease [5]. Cold exposure is known to increase blood pressure [6,7]. A review by Jevons et al. found sufficient evidence to recommend a threshold ambient temperature for health in the UK [8]. They advocated for a population wide threshold of 18˚C to minimise potential harm to both vulnerable and healthy portions of the population. However, under conditions of milder exposure for healthy individuals, the relationship between cold and morbidity is complex. Recent work has found evidence that mild cold exposure may moderately improve metabolic health [9]. In general, humans exhibit a wide range of adaptive physiological responses to cold [10] and emerging evidence suggests that cold adaptation could lead to a decreased risk of cardiovascular disease [11].

From a thermal comfort perspective, there is increased interest in indoor environments which do not provide static, isothermal conditions on a room/dwelling basis [12]. Presently, UK domestic temperatures tend to be controlled by gas central heating systems operated by a single thermostat [13]; UK offices are typically regulated by centrally controlled air supply systems (HVAC), which lack opportunities for personal control [14]. The deployment of personal comfort systems allows the user to tune their local environment to their personal preferences, against a background heating or cooling load which would provide minimal comfort by itself. This may take the form of heated seating, or air-flow control systems, which are provided at the user's workspace. Such systems have the potential to reduce heating demand considerably [15, 14]. Heterogeneous indoor microclimates provided by such systems could be deliberately constructed to introduce thermal asymmetries and local air movement [16]. As a result, the range of temperatures experienced throughout the day would increase, as well as offering opportunities for personal control [17].

This paper reports the findings of a novel research project which aimed to characterise the immediate thermal environment of a study participant using a wrist worn monitor. No program of study prior to this has sought to understand the variations of experienced temperature at the population level, although earlier findings from this study were reported previously [18].

In this study, we used data collected as part of the UK Biobank–a large on-going longitudinal health study of older UK adults [19]. Participants wore an Axivity AX3 wristband for a single week between June 2013 and December 2015, which recorded the experienced temperature and activity levels of the participant. Earlier work showed the experienced temperature to be a mix of ambient temperature and heat from the wrist [18]. The study design is cross-sectional in nature. Two models were constructed to understand how the standard deviation of experienced temperature, the thermal variety, is related to health outcomes. The experienced temperature was down-sampled to a 1-minute interval. Model 1 used self-reported health satisfaction as its primary independent variable of interest, while model 2 used diagnosed health conditions. The overall aim of this study was to contribute empirical evidence to our understanding of the relationship between health and the temperatures that people experience in daily life.

## Methods

With the exception of external temperature, all variables used in this study were collected as part of the UK Biobank. All data were fully anonymised by UK Biobank prior to being shared with the study authors. The thermal variety and activity level variables were derived from the Axivity AX3 wristband measurements using the cluster computing environment of University College London (UCL). The computational script was a modified version of one produced by Doherty et al. for their work on the AX3 monitor [20]. A calibration error by the Axivity manufacturers was discovered and corrected by the author. 103,707 files were processed, comprising 27TB of data in total. This processing produced a down-sampled timeseries at a 1-minute period. The temperature data were recorded at a period of between 1.1–1.3 seconds. The

accuracy of the AX3 temperature sensor is ±1˚C under standard operating conditions and the resolution is 0.3˚C [21]. The response time is on the order of industry standard temperature monitors (i.e. Onset's HOBO U12 Data Logger).

While the data were being processed, 4 participants withdrew from the study. UK Biobank provided the anonymised identification codes of these participants and their data were deleted. A minimum wear-time criterion of 90% was imposed (9072 minutes out of a total possible 10800 for the study week). Participants who conducted nightshift work were excluded, as were those who were diagnosed with dementia or Alzheimer's disease, as these may be associated with circadian disruption [22]. Participant's whose average activity was greater than 0.1g were also excluded, as were timeseries which exhibited clear sensor malfunction or substantial missing data. From these processed timeseries, thermal variety was calculated as the standard deviation of the temperature measured by the device during the week of wear.

## UK Biobank variables

The variables selected for inclusion were based on a pre-analysis plan for a previous portion of the present study [23]. The variables of activity level and thermal variety were both recorded using the AX3 device between June 2013 and December 2015. All other variables were collected at the time of initial assessment, between 2006 and 2010, and subsequent follow-up visits [19] with the exception of external temperature (see below). The variables of age, household size, body mass index and activity level were binned into appropriate categories to aid interpretability.

The variable $C_{EWD}$ (whether or not a participant had a condition associated with excess winter deaths) used in Model 2, was constructed from the UK Biobank data on diagnosed disease. $C_{EWD}$ was given the value 1 if participants had been diagnosed with either a respiratory disease (defined under the 10th iteration of the International Statistical Classification of Diseases and Related Health Problems (ICD-10) as codes J00 to J99) or a circulatory disease (ICD-10 codes I00 to I99). Alzheimer's disease and dementia (ICD-10 codes F01 –F03) are also related to excess winter deaths [24] but they were excluded due to the potential for circadian disruption as described above—only six participants in the study had such diseases.

## External temperature

For all participants the average local external temperature for the week in which the AX3 device was worn was calculated. The rounded (1 km) home location of each participant was matched to the corresponding grid square of NASA's MEERA-2 surface temperature dataset [25, 26]. The grid resolution was 0.625˚×0.5˚ (approximately 70×35 km).

## Model 1 (N = 37,730)

Model 1 was a linear model using thermal variety as the dependent variable, with the following independent variables: external temperature, age, health satisfaction, financial situation satisfaction, heating type, sex, ethnic background, household income, tenure type, accommodation type, household size, employment status, gas or solid-fuel cooking/heating, body mass index and activity level. The inclusion of self-reported health satisfaction was designed to capture the subjective sense of well-being and its association with thermal variety.

## Model 2 (N = 77,762)

Model 2 was a binomial linear model using a log link function between the dependent variable $C_{EWD}$ and the following independent variables: age, sex, ethnic background, household

income, tenure type, accommodation type, household size, employment status, gas or solid-fuel cooking/heating, body mass index, activity level and thermal variety. Model 2 did not include health satisfaction, financial situation satisfaction or heating type as these variables were only available for 37,770 participants and the model did not converge with this lower number of participants. The use of $C_{EWD}$ as the independent variable in model 2 was designed to triangulate any findings of model 1 in relation to health satisfaction.

## Results

In model 1, thermal variety was the main outcome variable in a multiple linear regression model against various demographic and health factors. The average external temperature for the week in which the thermal variety was recorded was also included, the relationship between them is shown in Fig 1, which shows that thermal variety is greater at the coldest times of the year.

For model 1, the full results are given in Tables 1 and 2. The clearest statistically significant results from model 1 show that thermal variety decreases with increasing age, increasing unhappiness with health satisfaction and increasing body mass index. Thermal variety increases with activity levels (S1 Fig). This is unlikely to be accounted for by physiological changes of wrist temperature alone. Studies of wrist temperature variation find the amplitude of variation to be around 1.0–1.5°C [27, 28], which would equate to at most a standard deviation of around 0.4°C. Those living in accommodation they own outright have higher thermal variety than those living in homes rented from the local authority. There were no significant differences as a function of household income.

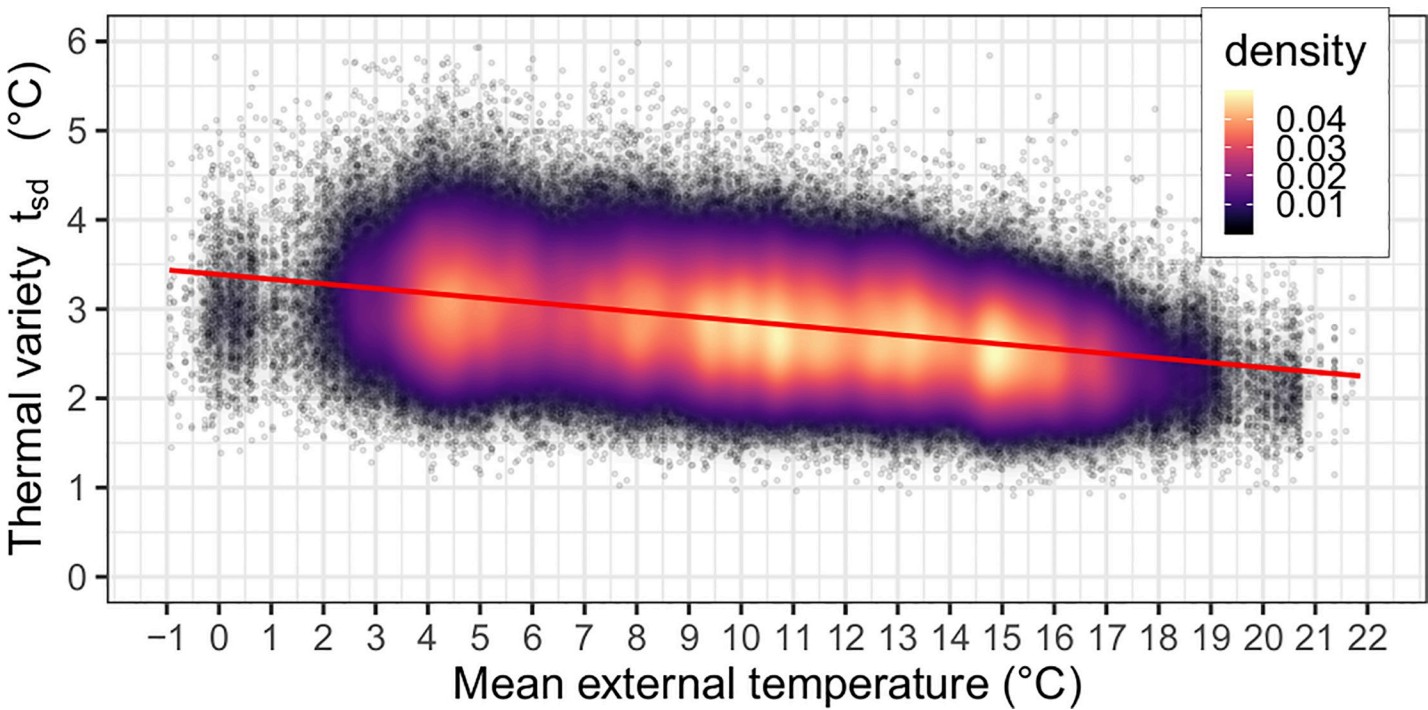

**Fig 1. The relationship between mean external temperature and thermal variety.** The relationship is approximately linear and shows higher thermal variety during the coldest periods of the year. Data were not sampled across a uniform distribution of external temperatures. The least square regression line is shown in red (β = -0.05, p<2x10$^{-16}$). Since 77,762 participants are plotted, the data is represented as a density cloud.

**Table 1. Multiple linear regression of thermal variety with demographic, building and health factors.** N = 77,762. $R^2$ = 0.24. Significance levels: * p<0.01, ** p<0.001, *** p<1x10$^{-9}$.

| Predictor (relative subcategory, N) | Sub-category (N) | $t_{sd}$°C |
|---|---|---|
| Intercept | - | 3.46 [3.43–3.48] *** |
| External temperature°C | - | -0.05 [-0.06 –-0.05] *** |
| Age (40–49, 6075) | 50–59 (21320) | -0.06 [-0.08 –-0.04] *** |
| | 60–69 (35407) | -0.10 [-0.12 –-0.08] *** |
| | 70–79 (14960) | -0.16 [-0.18 –-0.14] *** |
| Sex (Female, 43770) | Male (33992) | -0.05 [-0.06 –-0.04] *** |
| Ethnic background (White, 75365) | Mixed (398) | 0.07 [0.01–0.13] |
| | Asian (654) | -0.01 [-0.05–0.04] |
| | Black (582) | 0.09 [0.04–0.14] ** |
| | Chinese (157) | 0.11 [0.02–0.21] |
| | Other ethnic group (395) | 0.03 [-0.03–0.09] |
| | Do not know (20) | 0.08 [-0.18–0.35] |
| | Prefer not to answer (191) | -0.05 [-0.13–0.04] |
| Household Income (Less than £18,000, 10592) | £18,000 to £30,999, (17779) | -0.02 [-0.04 –-0.01] * |
| | £31,000 to £51,999 (20016) | -0.01 [-0.03–0.00] |
| | £52,000 to £100,000 (17021) | -0.01 [-0.03–0.01] |
| | Greater than £100,000 (4850) | -0.02 [-0.04–0.01] |
| | Prefer not to answer (5475) | -0.01 [-0.03–0.01] |
| | Do not know (2029) | -0.07 [-0.10 –-0.04] ** |
| Accommodation type (House/bungalow, 71554) | Flat (6058) | -0.07 [-0.09 –-0.05] *** |
| | Temporary (54) | 0.02 [-0.14–0.18] |
| | None of above (83) | -0.05 [-0.18–0.08] |
| | Prefer not to answer (13) | -0.17 [-0.51–0.17] |
| Tenure type (Own outright, 44537) | Mortgage (28498) | -0.05 [-0.07 –-0.04] *** |
| | Rent Local Authority (2096) | -0.16 [-0.18 –-0.13] *** |
| | Rent private (1497) | -0.04 [-0.07 –-0.01] |
| | Shared (174) | -0.07 [-0.16–0.02] |
| | Rent free (469) | -0.09 [-0.15 –-0.04] |
| | None of above (276) | -0.07 [-0.14–0.00] |
| | Prefer not to answer (215) | -0.01 [-0.09–0.08] |
| Household size (single occupant, 12854) | Two (37905) | -0.04 [-0.05 –-0.02] ** |
| | Three (12141) | -0.05 [-0.06 –-0.03] *** |
| | Four or more (14862) | -0.03 [-0.05 –-0.01] *** |

(*Continued*)

**Table 1.** (Continued)

| Predictor (relative subcategory, N) | Sub-category (N) | $t_{sd}$°C |
|---|---|---|
| Employment status (In paid employment or self-employed, 39797) | Retired (27472) | 0.03 [-0.03–0.09] |
| | Looking after home/family (3235) | 0.03 [-0.09–0.15] |
| | Unable to work, sickness/disability (1411) | 0.01 [0.00–0.02] |
| | Unemployed (901) | 0.02 [-0.00–0.04] |
| | Doing unpaid or voluntary work (3759) | -0.10 [-0.13 –-0.06] ** |
| | Full/ part-time student (738) | -0.02 [-0.06–0.02] |
| | None of the above (350) | 0.04 [0.01–0.06] ** |
| | Prefer not to answer (99) | 0.04 [-0.00–0.08] |
| Fuel type (Gas hob or gas cooker, 28957) | Gas fire (6379) | 0.01 [-0.00–0.03] |
| | Open solid fuel fire (2335) | 0.12 [0.09–0.14] *** |
| | Gas hob & Gas fire (20188) | 0.01 [-0.00–0.02] |
| | Gas hob & Open solid fuel fire (4481) | 0.09 [0.07–0.11] *** |
| | Gas fire & Open solid fuel fire (195) | 0.21 [0.12–0.29] ** |
| | Gas hob & Gas fire & Open fire (956) | 0.08 [0.04–0.12] ** |
| | None of the above (14221) | -0.01 [-0.02–0.00] |
| | Prefer not to answer (37) | -0.21 [-0.41 –-0.01] |
| | Do not know (13) | -0.18 [-0.50–0.15] |
| Body Mass Index (Normal, 30562) | Underweight (477) | 0.11 [0.06–0.17] ** |
| | Overweight (45722) | -0.18 [-0.19 –-0.18] *** |
| | Obese (1001) | -0.37 [-0.41 –-0.34] *** |
| Activity level recorded (1st quintile, 15463) | 2nd quintile (15567) | 0.14 [0.13–0.16] *** |
| | 3rd quintile (15567) | 0.24 [0.22–0.25] *** |
| | 4th quintile (15578) | 0.33 [0.31–0.34] *** |
| | 5th quintile (15587) | 0.50 [0.49–0.51] *** |

Model 2 was constructed to understand how conditions associated with excess winter deaths ($C_{EWD}$) are related to the thermal variety of the participant. $C_{EWD}$ was constructed as a binary variable and denoted whether a participant had been diagnosed with cardiovascular or respiratory conditions. A binomial regression model of the relationship between $C_{EWD}$, thermal variety and potentially confounding demographic factors was produced. These findings are given in Table 3.

Model 2 found that the risk of $C_{EWD}$ decrease with increasing thermal variety, activity and income. Risk of $C_{EWD}$ increase as a function of age and body mass index. Being unable to work because of sickness or disability also showed a strong increased risk of $C_{EWD}$, as might be expected. Renting from the local authority had a strong increased risk of $C_{EWD}$ over owning a home outright; having a mortgage showed moderate increased risk. Unlike model 1 there was a clear effect as a function of income–a higher household income was associated with decreased risk of $C_{EWD}$ across all income brackets. The absence of a significant relationship for household income in model 1 is addressed in the supporting information section.

It is important to stress that the associations highlighted by both models in this study do not necessarily point to causal mechanisms. It is possible that those who have health conditions are less able to access, or seek to avoid, wide thermal ranges.

**Table 2. Additional variables for the regression given in Table 1.** These variables were only available for N = 37,730 participants. $R^2 = 0.24$. Significance levels: * $p<0.01$, ** $p<0.001$, *** $p<1 \times 10^{-9}$.

| Predictor (relative subcategory, N) | Sub-category (N) | $t_{sd}$ |
|---|---|---|
| Health satisfaction (Extremely happy, 2230) | Very happy (13771) | -0.04 [-0.07 –-0.01] * |
| | Moderately happy (17767) | -0.10 [-0.12 –-0.07] *** |
| | Moderately unhappy (2955) | -0.15 [-0.19 –-0.12] *** |
| | Very unhappy (661) | -0.16 [-0.21 –-0.11] ** |
| | Extremely unhappy (249) | -0.15 [-0.23 –-0.07] ** |
| | Prefer not to answer (10) | 0.05 [-0.32–0.42] |
| | Do not know (87) | -0.04 [-0.17–0.08] |
| Financial situation satisfaction (Extremely happy, 3808) | Very happy (14498) | 0.01 [-0.01–0.04] |
| | Moderately happy (15732) | 0.01 [-0.01–0.03] |
| | Moderately unhappy (2473) | -0.02 [-0.05–0.02] |
| | Very unhappy (737) | -0.07 [-0.12 –-0.02] * |
| | Extremely unhappy (369) | -0.06 [-0.13–0.00] |
| | Prefer not to answer (57) | -0.09 [-0.25–0.07] |
| | Do not know (56) | -0.12 [-0.28–0.03] |
| Heating Type (Gas central heating, 34999) | Electric storage heaters (798) | -0.01 [-0.05–0.03] |
| | Oil (kerosene) central heating (979) | 0.09 [0.05–0.13] ** |
| | Portable gas or paraffin heaters (10) | 0.17 [-0.20–0.54] |
| | Solid fuel central heating (128) | 0.09 [-0.01–0.20] |
| | Open fire without central heating (109) | -0.02 [-0.14–0.09] |
| | Three heating types (5) | -0.17 [-0.69–0.35] |
| | None of the above (676) | -0.01 [-0.05–0.04] |
| | Prefer not to answer (15) | -0.19 [-0.53–0.16] |
| | Do not know (11) | -0.17 [-0.52–0.19] |

## Discussion

A conceptual representation of the above findings, informed by the literature as a whole, is given in Fig 2. This shows that, for healthy individuals, a wider range of experienced temperature is found. This is evidenced by the findings of both models in this study; individuals with greater health satisfaction have larger thermal variety, and the risk of having a condition associated with excess winter deaths is lower for each degree increase of thermal variety. This is equivalent to a larger range of temperatures which are not harmful for healthy individuals. For individuals who are less healthy, the range of healthy temperatures is narrower. Ultimately this effect likely contributes to the greater mortality levels in winter in temperate climates i.e. those living with respiratory or cardiovascular diseases experience higher risks in extreme temperatures than those without such conditions.

Differences in experienced temperature, and thermal variety, are associated with a number of demographic and housing factors. In terms of Fig 2, reducing harmful exposure necessitates modifying internal temperatures in underheated dwellings. Thermal variety can also be framed as an issue of flexibility justice [29]. This concept suggests that the ability to be flexible in daily life constitutes a form of capital, which is unevenly distributed in society. In many future energy scenarios flexibility will be increasingly valuable. The figure can therefore be interpreted from a flexibility justice perspective; those in good health can potentially tolerate increased thermal variety, whereas those in poor health might require a narrowing of their

**Table 3. Risk ratio of $C_{EWD}$ as a function of both $t_{sd}$ and other demographic, health and building factors.**
N = 77,762 Significance levels: * p<0.01, ** p<0.001, *** p<1x10$^{-9}$.

| Predictor (relative subcategory) | Sub-category | Risk ratio ($t_{sd}$) |
|---|---|---|
| $t_{sd}$ | - | 0.95 [0.93–0.98] ** |
| Age (40–49) | 50–59 | 1.48 [1.32–1.64] *** |
| | 60–69 | 2.10 [1.88–2.34] *** |
| | 70–79 | 2.70 [2.41–3.03] *** |
| Sex (Female) | Male | 1.52 [1.47–1.58] *** |
| Ethnic background (White) | Mixed | 0.99 [0.74–1.32] |
| | Asian or Asian British | 1.16 [0.97–1.38] |
| | Black or Black British | 0.90 [0.71–1.15] |
| | Chinese | 0.86 [0.51–1.44] |
| | Other ethnic group | 1.12 [0.88–1.44] |
| | Do not know | 1.17 [0.41–3.39] |
| | Prefer not to answer | 0.95 [0.68–1.33] |
| Household income per year (less than £18,000) | £18,000 to £30,999 | 0.91 [0.86–0.96] ** |
| | £31,000 to £51,999 | 0.80 [0.75–0.85] *** |
| | £52,000 to £100,000 | 0.72 [0.67–0.77] *** |
| | Greater than £100,000 | 0.64 [0.57–0.71] *** |
| | Prefer not to answer | 0.84 [0.77–0.91] ** |
| | Do not know | 0.92 [0.82–1.03] |
| Tenure type (Own outright) | None of above | 0.91 [0.66–1.26] |
| | Prefer not to answer | 1.08 [0.78–1.49] |
| | Mortgage | 1.07 [1.03–1.12] * |
| | Rent Local Authority | 1.22 [1.10–1.35] ** |
| | Rent private | 1.11 [0.98–1.26] |
| | Shared | 1.47 [1.07–2.01] |
| | Rent free | 1.04 [0.83–1.30] |
| Accommodation type (House or bungalow) | Flat | 0.97 [0.91–1.05] |
| | Temporary | 0.81 [0.41–1.62] |
| | None of above | 0.80 [0.45–1.41] |
| | Prefer not to answer | 1.18 [0.40–3.52] |
| Employment status (In paid/self-employment) | Retired | 1.06 [1.01–1.11] |
| | Looking after home and/or family | 0.96 [0.86–1.08] |
| | Unable to work due to sickness/disability | 1.82 [1.66–1.99] *** |
| | Unemployed | 0.85 [0.71–1.01] |
| | Doing unpaid or voluntary work | 1.06 [0.97–1.15] |
| | Full or part-time student | 1.04 [0.84–1.28] |
| | None of the above | 1.10 [0.85–1.41] |
| | Prefer not to answer | 0.76 [0.42–1.36] |
| Fuel type (Gas hob or gas cooker) | Open solid fuel fire | 1.01 [0.91–1.13] |
| | Gas hob & Gas Fire | 1.05 [1.00–1.09] |
| | Gas hob & solid fuel open fire | 0.92 [0.85–1.01] |
| | Gas fire & solid fuel open fire | 1.05 [0.74–1.50] |
| | Gas hob & Gas fire & solid fuel open fire | 0.98 [0.83–1.16] |
| | None of the above | 1.01 [0.96–1.06] |
| | Prefer not to say | 1.64 [0.95–2.85] |
| | Do not know | 1.52 [0.63–3.67] |

(*Continued*)

**Table 3.** (Continued)

| Predictor (relative subcategory) | Sub-category | Risk ratio ($t_{sd}$) |
|---|---|---|
| Body mass index (Normal) | Underweight | 1.00 [0.76–1.31] |
| | Overweight | 1.15 [1.11–1.20] *** |
| | Obese | 1.49 [1.32–1.68] *** |
| Activity level recorded (1st quintile) | 2nd quintile | 0.84 [0.80–0.89] *** |
| | 3rd quintile | 0.81 [0.77–0.86] *** |
| | 4th quintile | 0.77 [0.73–0.82] *** |
| | 5th quintile | 0.72 [0.68–0.77] *** |
| Household size (single) | Two | 1.13 [1.07–1.19] ** |
| | Three | 1.20 [1.12–1.29] ** |
| | Four or more | 1.18 [1.09–1.27] ** |

thermal variety to avoid harmful exposure. Since only associations are highlighted by the present study, these interpretations should be caveated by noting that causal relationships have not be revealed by this study.

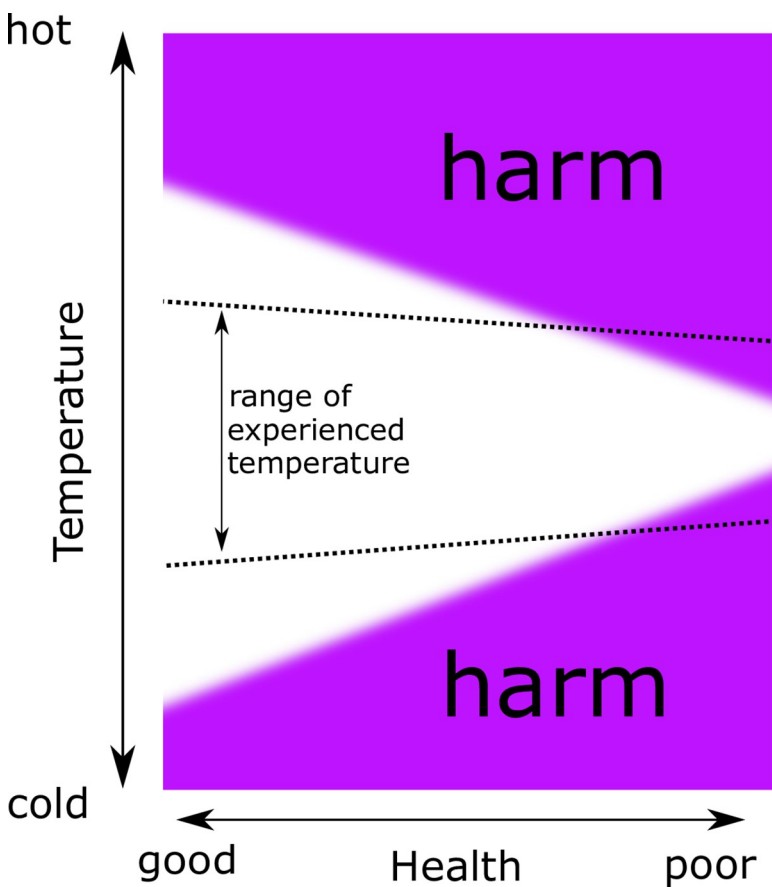

**Fig 2. A schematic summary of the results of this study (dotted black lines) and the conceptual structure of the broader literature.** Healthy individuals have a wider range of temperatures that are not harmful, and typically do not experience them. Individuals with poor health have a narrower range of experienced temperature, and are more likely to experience harmful thermal conditions, especially when living in poor housing which fails to guard against harmful temperature exposure. This harmful exposure is understood to contribute to the observed seasonal variation in mortality.

Low thermal variety, especially in winter, may also point to the problem of chronically low experienced temperature [30]. The data as a whole (see S2 Fig) show that lower mean temperatures are weakly associated with higher thermal variety, although this is most likely a seasonal effect. For fuel poor households that struggle to afford warmth due to a combination of low income, poor thermal dwelling performance and high energy costs, low thermal variety may be more likely. It is vital that attention is focused on at-risk populations who lack the means to avoid harmful cold exposure. Secondary health impacts associated with temperature such as mould growth and damp, which are more prevalent in underheated homes, are also a priority.

In the EU, space cooling remains uncommon in homes [1]. Since the vast majority of domestic heating systems there are fuelled by carbon intensive resources, allowing more thermal variation in dwellings occupied by healthy individuals could yield carbon emissions savings. From a policy standpoint, such a position is currently controversial given that government and health body recommendations typically avoid differentiating between thermal environments for healthy and unhealthy individuals. However, when coupled with the emerging evidence from the thermal comfort literature on the comfort potential of indoor environments which avoid thermal monotony, such a proposal has broader appeal. Heating reduction campaigns could be targeted at healthy, well-off and environmentally conscious portions of the population as a means of raising awareness of the climate impacts of $CO_2$ intensive heating. As economies of the Northern Hemisphere undergo the transition away from carbon, providing low-carbon comfort for those able to tolerate wider thermal variety would allow carbon intensive heating to be reserved for those most in need. Practically, this could take the form of health differentiated heating recommendations, moving away from a one size fits all approach, towards policy which focuses on health-related needs as well as also carbon emission reductions targets.

## Supporting information

**S1 Fig. The relationship between mean activity in mg and thermal variety.**
$t_{sd} = 0.02\bar{a} + 2.19$, where $\bar{a}$ is the mean recorded activity for the study week.
(TIF)

**S2 Fig. The relationship between mean experienced temperature and thermal variety.**
$t_{sd} = -0.1\bar{t} + 5.8$, where $\bar{t}$ is the mean experienced temperature.
(TIF)

## Acknowledgments

This research has been conducted using the UK Biobank Resource under Application Number 26284.

The authors are grateful to Raquel Alegre and Stuart Grieve of UCL's Research IT services for their assistance in processing 27TB of data, and to Ed Sharpe for suggesting the use of NASA's MEERA-2 dataset during the project's development.

## Author Contributions

**Conceptualization:** Harry R. Kennard, Gesche M. Huebner, David Shipworth.

**Data curation:** Harry R. Kennard.

**Formal analysis:** Harry R. Kennard.

**Funding acquisition:** Gesche M. Huebner, David Shipworth, Tadj Oreszczyn.

**Investigation:** Harry R. Kennard, Gesche M. Huebner, David Shipworth.

**Methodology:** Harry R. Kennard, Gesche M. Huebner, David Shipworth.

**Project administration:** Harry R. Kennard, Gesche M. Huebner, David Shipworth.

**Resources:** Gesche M. Huebner, David Shipworth.

**Software:** Harry R. Kennard.

**Supervision:** Gesche M. Huebner, David Shipworth.

**Validation:** Harry R. Kennard, Gesche M. Huebner, David Shipworth, Tadj Oreszczyn.

**Visualization:** Harry R. Kennard.

**Writing – original draft:** Harry R. Kennard.

**Writing – review & editing:** Harry R. Kennard, Gesche M. Huebner, David Shipworth, Tadj Oreszczyn.

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
