## [Decision Letter · Decision Letter 0]

12 May 2020

PONE-D-20-02222

The associations between thermal variety and health: implications for space heating energy use

PLOS ONE

Dear Dr Kennard,

Thank you for submitting your manuscript to PLOS ONE. After careful consideration, we feel that it has merit but does not fully meet PLOS ONE’s publication criteria as it currently stands. Therefore, we invite you to submit a revised version of the manuscript that addresses the points raised during the review process.

ACADEMIC EDITOR: 

Based on the comments made by the reviewers, I have a positive feeling that the paper can be accepted for publication once all comments are properly addressed.

Please check the English writing by further proofreading the text and remove any typos from the text.

Use proper citations and also make sure that you have asked permission for any figures already published in the literature. 

It would be more appealing if a paragraph is written on the contribution, gap and novelty of the present work in the "introduction" section.

We would appreciate receiving your revised manuscript by Jun 26 2020 11:59PM. To enhance the reproducibility of your results, we recommend that if applicable you deposit your laboratory protocols in protocols.io, where a protocol can be assigned its own identifier (DOI) such that it can be cited independently in the future. For instructions see: http://journals.plos.org/plosone/s/submission-guidelines#loc-laboratory-protocols

We look forward to receiving your revised manuscript.

Kind regards,

Mason Sarafraz

Academic Editor

PLOS ONE

2. In ethics statement in the manuscript and in the online submission form, please provide additional information about the patient records used in your retrospective study. Specifically, please ensure that you have discussed whether all data were fully anonymized before you accessed them and/or whether the IRB or ethics committee waived the requirement for informed consent. If patients provided informed written consent to have data from their medical records used in research, please include this information.

4. Please upload a copy of Supporting Information Figure 4 which you refer to in your text on page 16.

Reviewers' comments:

Reviewer's Responses to Questions

**Comments to the Author**

1. Is the manuscript technically sound, and do the data support the conclusions?

Reviewer #1: Yes

Reviewer #2: Yes

2. Has the statistical analysis been performed appropriately and rigorously? 

Reviewer #1: Yes

Reviewer #2: Yes

3. Have the authors made all data underlying the findings in their manuscript fully available?

Reviewer #1: No

Reviewer #2: Yes

4. Is the manuscript presented in an intelligible fashion and written in standard English?

Reviewer #1: Yes

Reviewer #2: Yes

5. Review Comments to the Author

Reviewer #1: Generally, it is scientifically sound and well written. Only minor revision may be required to clarify some ambiguities. You can find my detail comments from the attached 'Reviewer's Comment' word document.

Reviewer #2: The original research article presents a interesting topic pertaining to environmental health. The manuscript is written in a intelligent fashion although the methodology section can be further strengthened by describing the study design. There were some topographical and grammatical errors detected throughout the manuscript. A professional editing is suggested. and A few suggestions are provided of relevant literature which can be incorporated in the introduction and discussion section.

Houghton A, Castillo-Salgado C. Associations between Green Building Design Strategies and Community Health Resilience to Extreme Heat Events: A Systematic Review of the Evidence. International journal of environmental research and public health. 2019 Jan;16(4):663.

Tsoulou I, Andrews CJ, He R, Mainelis G, Senick J. Summertime thermal conditions and senior resident behaviors in public housing: A case study in Elizabeth, NJ, USA. Building and Environment. 2020 Jan 15;168:106411.

Asumadu-Sakyi AB, Barnett AG, Thai PK, Jayaratne ER, Miller W, Thompson MH, Rahman MM, Morawska L. Determination of the association between indoor and outdoor temperature in selected houses and its application: A pilot study. Advances in Building Energy Research. 2019 Apr 23:1-35.

6. PLOS authors have the option to publish the peer review history of their article (what does this mean?). If published, this will include your full peer review and any attached files.

Reviewer #1: Yes: Serebe Gebrie

Reviewer #2: No

---

## [Author Response · Author response to Decision Letter 0]

16 Jun 2020

Dear Mason Sarafraz, editors and reviewers,

The authors would like to thank the reviewers and the editor for their positive review and helpful comments, as well as taking the time to carry the review.

Before addressing specific reviewer and editorial comments, it is helpful to clarify the data availability for this study. The data for this study were derived from the UK Biobank, which places restrictions on data sharing, but allows any bonified researcher to apply for access. Therefore, the authors would like to amend the data availability statement to the following:

“Restrictions limit the direct sharing of the data used in this study. However, the data used in this study are available to any bona fide researchers following application to a third party, UK Biobank at www.ukbiobank.ac.uk. All data were fully anonymised by UK Biobank prior to being shared with the study authors”

For precedence regarding the use of UK Biobank data in Plos One studies, please see, for example: Lyall et al 2016. https://journals.plos.org/plosone/article?id=10.1371/journal.pone.0154222. The authors hope that this clarification addresses both the data-sharing and ethics queries raised.

The submission has been proofread to ensure no spelling errors remain. All submission figures have been checked and processed using the PACE provided. An additional reference pertaining to personal thermal control in office systems added. 

The specific editorial and reviewer comments are addressed below:

Editorial comments: 

Comment Response

It would be more appealing if a paragraph is written on the contribution, gap and novelty of the present work in the "introduction" section. An extra framing paragraph has been included (lines 69-73)

Please upload a copy of Supporting Information Figure 4 which you refer to in your text on page 16. Figure 4 was intended to refer to figure S2. The text has been updated accordingly. (line 277)

Reviewer #2: 

Comment Response

“The original research article presents a interesting topic pertaining to environmental health. The manuscript is written in a intelligent fashion although the methodology section can be further strengthened by describing the study design. There were some topographical and grammatical errors detected throughout the manuscript. A professional editing is suggested. and A few suggestions are provided of relevant literature which can be incorporated in the introduction and discussion section.” The study design has been further expounded on lines 79-80.

Typographical errors have been corrected.

The authors are grateful for the suggested additional literature. However, the paper already includes significant recent review articles (refs 4-11) related to the topic in question. The literature suggestion pertaining to green building design is too specific for inclusion here. The second and third suggested studies are a case study and a pilot study respectively, and so also lack general relevance required for inclusion in this paper. 

Serebe Gebrie comments: 

Comment Response

(Page 4, L61-64) – It shows thermal asymmetries and local air movement would offer opportunities for personal control. Would you explain a little about how temperature is controlled or regulated in the UK? what are the major problems with the current system? A description of typical temperature control systems in both domestic and office settings has been added (lines 55-58) with relevant references.

In this study, you have used two models. What is the rationale to use these two models? It seems model 2 is more reliable than model 1. Self-reported health satisfaction could be more subjective and has a higher risk of bias. It would be clear if you could add the reason why you have used these two models. Lines 69-73 have been added, including relevant references, to address this comment. 

Lines 143-144 and 152-153 have been added to clarify the choice of models. The use of self-reported health status is essential if subjective assessments of well-being are to be captured. It is not unreliable per se, since it aims to capture subjective well-being. 

In this paper, there are some abbreviations and subject-specific words. Eg. ICD, HOBO, UCL’s cluster. I suggest using abbreviations and operational definitions. Otherwise, you may define or write in full text when first use. All subject specific abbreviations have been defined at first use in the text, and/or clarified where required.

(Page 5, L91&92) – “While the data were being processed, 4 participants withdrew from the study, and their data were deleted.”. This sentence gives me a little confusion. Have you used primary data directly from participants? If so, you need to add ‘Ethical consideration’. Per the updated data availability statement above, data were provided anonymised by the UK Biobank. The text has been amended to clarify the deletion process, lines 101-102.

Results: The reader may expect to see why household income was not associated with health risks in model 1, but it did in model 2. What could be the reason for this variation? Please include it in your discussion. A comment in the main text directing the reader to a statement in the supporting information has been added (lines 449-454).

On behalf of the authors, we hope that this resubmission satisfies the requirements of the journal, editors and reviewers.

With best wishes,

Dr Harry Kennard

---

## [Editor Report · Decision Letter 1]

30 Jun 2020

The associations between thermal variety and health: implications for space heating energy use

PONE-D-20-02222R1

Dear Dr. Kennard,

We’re pleased to inform you that your manuscript has been judged scientifically suitable for publication and will be formally accepted for publication once it meets all outstanding technical requirements.

Kind regards,

Dr Mason Sarafraz

Academic Editor

PLOS ONE
---

## [Editor Report · Acceptance letter]

1 Jul 2020

PONE-D-20-02222R1 

The associations between thermal variety and health: implications for space heating energy use 

Dear Dr. Kennard:

I'm pleased to inform you that your manuscript has been deemed suitable for publication in PLOS ONE. Congratulations! Your manuscript is now with our production department. 

Kind regards, 

on behalf of

Dr. Mason Sarafraz 

Academic Editor

PLOS ONE